# Real-Time In-Vehicle Air Quality Monitoring System Using Machine Learning Prediction Algorithm

**DOI:** 10.3390/s21154956

**Published:** 2021-07-21

**Authors:** Chew Cheik Goh, Latifah Munirah Kamarudin, Ammar Zakaria, Hiromitsu Nishizaki, Nuraminah Ramli, Xiaoyang Mao, Syed Muhammad Mamduh Syed Zakaria, Ericson Kanagaraj, Abdul Syafiq Abdull Sukor, Md. Fauzan Elham

**Affiliations:** 1Faculty of Electronic Engineering Technology, Universiti Malaysia Perlis (UniMAP), Arau 02600, Malaysia; ccgoh@studentmail.unimap.edu.my (C.C.G.); nuraminah@unimap.edu.my (N.R.); smmamduh@unimap.edu.my (S.M.M.S.Z.); ericsonkanagaraj@gmail.com (E.K.); 2Advanced Sensor Technology, Centre of Excellence (CEASTech), Universiti Malaysia Perlis (UniMAP), Arau 02600, Malaysia; ammarzakaria@unimap.edu.my (A.Z.); abdulsyafiq@unimap.edu.my (A.S.A.S.); 3Faculty of Electrical Engineering Technology, Universiti Malaysia Perlis (UniMAP), Arau 02600, Malaysia; 4Graduate Faculty of Interdisciplinary Research, University of Yamanashi, 4-3-11 Takeda, Kofu, Yamanashi 400-8511, Japan; hnishi@yamanashi.ac.jp (H.N.); mao@yamanashi.ac.jp (X.M.); 5Selangor Industrial Corporation Sdn Bhd, Seksyen 14, Shah Alam 40000, Malaysia; fauzan@sic.com.my

**Keywords:** internet of things (IoT), machine learning prediction, in-vehicle air quality, smart mobility, smart city

## Abstract

This paper presents the development of a real-time cloud-based in-vehicle air quality monitoring system that enables the prediction of the current and future cabin air quality. The designed system provides predictive analytics using machine learning algorithms that can measure the drivers’ drowsiness and fatigue based on the air quality presented in the cabin car. It consists of five sensors that measure the level of CO_2_, particulate matter, vehicle speed, temperature, and humidity. Data from these sensors were collected in real-time from the vehicle cabin and stored in the cloud database. A predictive model using multilayer perceptron, support vector regression, and linear regression was developed to analyze the data and predict the future condition of in-vehicle air quality. The performance of these models was evaluated using the Root Mean Square Error, Mean Squared Error, Mean Absolute Error, and coefficient of determination (*R*^2^). The results showed that the support vector regression achieved excellent performance with the highest linearity between the predicted and actual data with an *R*^2^ of 0.9981.

## 1. Introduction

One of the main aims of smart cities is to reduce the fatalities and injuries due to traffic accidents. According to transport statistics in Malaysia, the total vehicles that are involved in road accidents increased yearly from 2008 to 2017. In 2017, the total road accidents reported were 533,875 cases and the total casualties and damages caused by traffic accidents were 16,589 cases [1]. The Royal Malaysia Police has stated that the leading causes of a road crash are drivers in fatigued conditions and distracted drivers [2]. The American Automobile Association (AAA) estimates that one out of every six deadly traffic accidents as well as one out of eight crashes requiring hospitalization is due to drowsy drivers [3]. In fact, the air inside the vehicle cabin has a significant impact on the cognitive capability of the occupants without noticeable discomfort that would put them on alert [4].

Most indoor air quality studies are focused on the inside of a building. The main components of indoor air contamination are carbon monoxide (CO), formaldehyde, ozone (O_3_), total volatile organic compounds (TVOC), and particulate matter (PM) which can highly affect human health [5]. A straightforward method of mitigating the hazardous gases is by closing all windows and doors to prevent the pollutants from the outdoors.

Furthermore, a similar indoor environment such as the vehicle cabin that has been equipped with a heater, ventilation, and air conditioning system (HVAC) can be categorized as an indoor space. HVAC systems use the recirculation mode (RC) that could mitigate the penetration of pollutants such as particulate matter and hazardous gases from the vehicle’s exhaust system [6,7]. Nonetheless, the human occupant inhales the oxygen then replaces it with carbon dioxide (CO_2_) which acts as contamination known as the human bio-effluent. The high concentration of CO_2_ reduces human cognitive ability, causes drowsiness, dizziness, and fatigue [8,9]. These are dangerous consequences to the occupants as well as potentially to other road users. Moreover, the vehicle speed might affect the gas concentration inside the vehicle cabin [10]. The undesirable effect of this condition is not limited to gas concentration only. The particle count (PM_2.5_ and PM_10_) circulating within the air can also affect the occupant’s health status [11]. The diameter of dust smaller than 2.5 µm can impact heart and lung health if inhaled by the occupants [12].

Thus, there is a clear need to perform an early prediction of the in-vehicle air quality which can be used to alert the occupants before the air quality becomes worse and affects the driver’s health condition while they are driving. Most of the previous studies only focused on classifying the hazardous gasses without having the ability to predict the future condition [13,14]. Furthermore, most studies are limited to a few hazardous gasses such as carbon dioxide (CO_2_) and oxygen (O_2_). In this respect, there are several methods from machine learning (ML) techniques such as artificial neural network (ANN) and regression algorithms that are applicable for air quality predictions. Furthermore, since the current reading of the air quality data depends on previous data, a time-sequence supervised learning air quality data can be used as the input structure [15].

This paper presents the design and development of low-cost sensor hardware for an in-vehicle air quality monitoring system with cloud-based storage and prediction on the current and future air quality. ML prediction methods are developed using several approaches such as multi-layer perceptron (MLP), support vector regression (SVR), and linear regression algorithms. These algorithms are compared to determine which is the best model by considering six inputs including the vehicle speed, CO_2_, temperature, humidity, PM_2.5_, and PM_10_. The contribution of this paper is the prediction system that includes the development of sensor hardware and cloud-based predictive analysis for an in-vehicle air quality monitoring system. The system is essential for future smart cities and smart mobility applications which can help to reduce fatality and injuries due to road accidents.

This paper is organized as follows. Section 2 discusses the related studies on air quality and prediction applications using ML algorithms. Section 3 describes the system development of an in-vehicle air quality monitoring system. Section 4 introduces the methods of collecting data. Section 5 shows the procedures of the data processing and Section 6 presents the proposed ML methods applied to the in-vehicle air quality data and the process flow of the ML analysis. The predicted result is then discussed in Section 7. Finally, the last section of this paper is the conclusion of the research.

## 2. Related Works

The literature studies have shown that humans spend up to 70%–90% of the time inside an environment with closed air circulation daily, including vehicle cabins [16,17]. The studies have shown that the air quality inside the vehicle cabin possibly contains polluted air [18,19,20]. They have also determined existing hazardous gases inside the cabin such as VOC, CO, CO_2_, nitrogen dioxide (NO_2_), sulphur dioxide (SO_2_), and other pollutants. What is worse is that the concentration level of those gases might be higher than the standards established by the World Health Organization (WHO) and other governmental health organizations. The effect might cause occupants to experience immediate health issues, including impaired vision and coordination, nose and throat irritations, headaches, dizziness, drowsiness, and fatigue to the occupants [8]. The combinations of these effects on the occupants’ health are not ideal for operating a vehicle.

The fresh air mode of the HVAC system triggered inside a vehicle can introduce air pollution from the outside environment such as: PM, NO_2_, SO_2_, and CO into the vehicle cabin. This can happen regularly especially in the urban and industrial areas. The RC mode significantly helps to reduce air pollution by circulating the air inside the vehicle cabin and increasing the passengers’ comfort experience. However, the RC mode can build up the CO_2_ concentration and accumulate rapidly due to the existence of the passengers.

Moreover, a statement has been declared by the Malaysian industry code of practice on the indoor environment that the CO_2_ concentration limit should not exceed 1000 ppm at any time. One study found that if the concentration of CO_2_ reached 2500 ppm in a room the size of 50.78 m^3^, the occupants’ decision-making capability on primary activity, initiative, information usage, breadth of approach, and basic strategy fall into a range of marginal and dysfunctional [16]. An average sedan vehicle’s interior space is around 2.72 m^3^ which is 18 times smaller than the experiment environment. A high CO_2_ concentration will reduce the O_2_ concentration and can cause permanent damage to organs, including the brain and heart [21].

In this paper, we focus entirely on the RC ventilation mode to operate inside the vehicle cabin. The primary pollutant source inside a cabin is the occupant. So, the CO_2_ concentration level is the main parameter to be observed as well as the PM concentration level. The fact of the matter is that different countries and organizations have different standards of the air quality index, even for the same type of pollutant. Some critical pollutants such as SO_2_ are not even taken into consideration in the air quality index for certain countries [22]. In addition, no established standard has showed the breakpoint concentration that is specifically for the in-vehicle air quality environment. Hence, Table 1 shows the combined in-vehicle air quality index breakpoints of the Environmental Protection Agency (EPA) standard, indoor air quality guideline in Malaysia [23], Occupational Safety and Health Administration (OSHA) [24], and the experiment that had been done by [16].

The US EPA has introduced the individual pollutant index, also known as the air quality index (AQI) as in Equation (1). The AQI acts as an indicator of reporting the air quality of the targeted environment. Equation (1) calculates each observed parameter in a time series. The highest individual index among other air parameters for pollutants will stand as the air quality of the vehicle’s cabin.
(1)$$I_{p}=\frac{I_{Hi}-I_{Lo}}{BP_{Hi}-BP_{Lo}}\;\left(C_{p}-BP_{Lo}\right)+I_{Lo}$$
where, $I_{p}$ = index for pollutant p$C_{p}$ = the rounded concentration of pollutant p$BP_{Hi}$ = the breakpoint that is greater than or equal to $C_{p}$$BP_{Lo}$ = the breakpoint that is less than or equal to $C_{p}$$I_{Hi}$ = the AQI value corresponding to $BP_{Hi}$$I_{Lo}$ = the AQI value corresponding to $BP_{Lo}$

With respect to prediction systems, artificial intelligence algorithms are widely used in smart city applications for classification prediction and regression prediction such as human activity classification [26,27], transportation [28], and air quality prediction [29,30,31,32]. In [33] the authors applied the ML algorithms to predict the air quality by using the data from 750 observations with 0.95 accuracy and their prediction was successful. [34] focused on predicting air pollution in Canada using an MLP, and the prediction model performed on PM_2.5_ had 4.5 of *MAE*.

Meanwhile, [35] have addressed the challenges in real-time air quality predictions, namely, the aspect of realistic real-time air quality monitoring devices, online systems, and predictive models in a review paper perspective. The real-time air quality system should provide an online user interface that allows the user to observe the air quality from anywhere. The support vector regression (SVR) is one of the most successful prediction models with a low root mean squared error (*RMSE*) (0.939) in forecasting the air quality in Japan [36,37] collected different types of gases and sent them to the cloud database. However, the authors addressed that the ML is used only for sensor calibration, not for in-vehicle air quality prediction. All these research studies are focused on indoor or outdoor air quality prediction, but not targeted for the air quality inside the vehicle cabin.

The research gap of this work is implementing the classification of the in-vehicle air quality together with the prediction of future conditions to monitor drivers’ dizziness and fatigue while they are driving. Hence, this work will be focused on system development and regression prediction for an in-vehicle air quality system.

## 3. System Design and Development

Based on the related works, the essential element to integrate a gas sensor into the hardware system is the type of sensor selection and targeted gas [38,39,40,41,42]. CO_2_ was found to be the most critical pollutant in the vehicle cabin with the RC ventilation mode. Moreover, the CO_2_ concentration is affected by the speed of the vehicle [43,44]. Thus, an integrated in-vehicle air quality monitoring system was developed for this work. The developed system has an integrated GPS tracking device as well as a CO_2_ gas sensor. Additional sensors such as particulate matter, temperature, and humidity are also embedded into the system. The overall system architecture of the in-vehicle air quality is illustrated in Figure 1. The system design is separated into four parts which are hardware development (device node), cloud database, software development (user interface), and an in-vehicle air quality prediction model. An IoT-oriented transportation system is applied in this system by connecting the device node to the internet in order to push real-time data into the cloud database [45].

The SIM808 (GPRS/GSM) communication module was chosen for this application. It not only provides better wireless regional coverage for up to 70 km but also has the feature to provide GPS coordinates. The raw GPS signal is cascaded with additional information which needs to be removed before storing only the latitudinal and longitudinal information. The microprocessor begins by initializing all the peripherals and sensors on the device node. Thirty seconds of initialization time is given to ensure all the sensors have been initialized properly. After initialization, a connection will be established between the device node and the cloud database by using the Message Queuing Telemetry Transport (MQTT) messaging protocol. Once the MQTT protocol connection has been established, the microprocessor begins gathering all the sensor data. Finally, the sensor data is aggregated into the buffer and encapsulated into the MQTT protocol format, and published into the cloud database using the brute force method. If the publishing is unsuccessful, the microcontroller checks the network of the MQTT connection and continues the sensor sampling process.

Furthermore, the data will be processed and sorted in the cloud database. Each of the device nodes is assigned with a unique identifier (ID) to avoid mis-location of the data entry. Then, a database handler is designed to discard the distorted data entry and invalid ID. For example, some published data may contain unidentified ASCII characters. Meanwhile, a web page and a mobile application are developed, which is capable of viewing the real-time data of the in-vehicle air quality status. The visualization is for the convenience of the user to understand and learn the patterns of the in-vehicle air quality. The data is illustrated in the form of Google Maps. There are several features available in the interface such as real-time view, data export, playback of the daily route, and view of historical data according to date.

The primary power source for the sensor device node is obtained from the in-car charger. The voltage range of a car battery is from 11.9V to 14.8V, where most of the time it is in the fluctuation mode. Therefore, several stages of voltage step-down are necessary due to the different operating voltages of the sensors. Figure 2 shows the final design of the sensor device baseboard with the specific voltage requirement and interface of the device node labelled. As for the sensor validation, the gas sensor data have been calibrated and verified with the established portable gas sensor device with the model Aeroqual, Series-500.

## 4. Real-Time Data Collections

In the preliminary study, the fresh air (FA) and RC ventilation mode are selected in order to observe the gas compositions inside the vehicle cabin. The eight gas parameters selected to be sampled inside the cabin are O_3_, CO, VOC, NO_2_, SO_2_, CO_2_, PM_2.5_, and PM_10_. The time taken for one set of data collection is forty minutes and will be repeated sixty times for each experiment. There are two types of cars used in this experiment that represents the indoor vehicle environment: the Nissan Grand Livina and the Toyota Vios. The experiment is conducted under an average vehicle speed of about 70 km/h.

Figure 3 shows the data collection of the common hazardous gases inside the vehicle cabin with the FA mode. For the FA mode, the quality of air inside the vehicle cabin is highly dependent on the outdoor air quality. Results show that most of the parameters observed exceeded the recommended limit established by the DOSH standard. Only two parameters—CO_2_ and SO_2_—are below the recommended limit. In the FA mode, the CO_2_ gas detected in the vehicle cabin was in the range of 600–900 ppm and the data collection showed a similar range with the outdoor air. It can be assumed that the low SO_2_ obtained is due to the experiment routes performed outside the petroleum refineries, chemical manufacturing industries, mineral ore processing plants, and power station areas. Meanwhile, the RC ventilation mode presented lesser hazardous gases existing inside the vehicle cabin. The only three parameters, which are CO_2_, PM_2.5_, and PM_10_ exceeded the recommended limit with reference to the DOSH standard. The air quality inside the vehicle cabin was not affected by the outdoor environment.

After finishing the preliminary experiment, we identified the essential observation parameters for further investigation. Next, the experiment is conducted on a real-time traffic basis. The experiment is performed for two months entirely under RC ventilation mode conditions. The travelling time can be separated into three slots, which are morning (06:00–08:00), afternoon (11:20–13:30), and evening (16:00–18:00). Travelling distance in June 2019 is 2306 km for 14 days, and July 2019 is 2494 km for 19 days. The daily average travelling distance is approximately 164.7 km, as shown in Figure 4. The minimum occupant is one and the maximum occupants are five. The experiment vehicle is a sedan car type with 2.75 m^3^ of space. Figure 5 shows the daily travelled path in this experiment. On the other hand, Table 2 shows the size of the data samples that have been collected throughout the experiments. Time series data in the air quality system has a parameter dedicated to counting the number of packets received by the cloud database. Each time the device is powered up, the count will be set as one and increases according to the subsequent data packet. Thus, the section data can be sorted using the count parameter. The average acquisition time for each data is 4 s. Once the data entered into the cloud database is completed, the vehicle speed is computed using the latitude, longitude, and time data. Then, the data labelling for the ML algorithms is generated. The data labelling is used to help the algorithms to train better and produce reliable results.

Figure 5 shows the raw CO_2_ sensor data on three different days for the morning slot. The fluctuation in the graph is due to the variation of the vehicle’s speed as well as the number of occupants in the car. The graph shows that the CO_2_ concentration level is higher than the recommended level by the Department of Occupational Safety and Health (DOSH). Other parameters of the morning slot such as vehicle speed, temperature, humidity, CO_2_, PM_2.5_, and PM_10_ are illustrated in Figure 6.

## 5. Data Processing for Time Series Data

This section will briefly introduce the flow of preparing the real-time data, labelling the data, and performing the data normalization procedure. This study uses time-series data, which is an ordered sequence data. The time interval between the data point is continuous and each time unit observation has at most, one data point.

The real-time sensor data might have data errors such as sensor error and outlier data prone to a false trend. The data preprocessing method is introduced to reduce the training complexity and to increase the accuracy while feeding the data into the prediction algorithms. The next step of the data preparation is labelling the sensor data. After completing the labelling process, the data will be run through a series of ML experiments to figure out the most compatible ML parameters for the air quality system. There are a total of six input data used for the ML, which are CO_2_, PM_2.5_, PM_10_, vehicle speed, temperature, and humidity.

### 5.1. Data Preprocessing

The raw data of the sensor is collected without a filtering process. The filtering process is implemented in the cloud rather than on the embedded device to reduce the complexity in the embedded system. The common data errors of the real-time monitoring application as expected are outliers and data missing [46]. Three types of common data pre-processing are: filling the not-a-number (NAN) data into zero, dropping the NAN data, or data interpolation before feeding the data into the ML algorithms. In this research, data interpolation is conducted using the nearest-neighbour method. This method is suitable for datasets that have missing values or outlier conditions [47]. Equation (2) shows the nearest-neighbour mathematical equations. When the outlier occurs at the position $x_{i}$, the value of the closest known neighbour is used to replace the outlier value. There are four states of different formulas that are used in this method. If the position of $x_{i}$ is greater than 5, the average of the five previous data will be used to replace the outlier position. When the outlier position is less than five, an average value will be used by taking as much historical data that it has. The reason for taking previous data and not using the future data is because the data collection is in real-time in time-series form.
(2)$$x_{i}=\{\begin{array}{c} \frac{x_{i-1}+x_{i-2}}{2}\;,\;if\;i=2;\\ \frac{x_{i-1}+x_{i-2}+x_{i-3}}{3}\;,\;if\;i=3;\\ \frac{x_{i-1}+x_{i-2}+x_{i-3}+x_{i-4}}{4}\;,\;if\;i=4;\\ \frac{x_{i-1}+x_{i-2}+x_{i-3}+x_{i-4}+x_{i-5}}{5}\;,\;\;otherwise \end{array}$$
where as $x_{i}$ is the outlier value.

### 5.2. Data Labelling

Before feeding the sensor data into the prediction algorithm, a set of data should be labelled as the output in supervised machine learning. In fact, the air quality index (AQI) is an index approach to categorize the quality of the air in a specific environment. The AQI is usually separated into a few ranges and each range is assigned a color code as well as a description. It provides a public health advisor for each range [48]. The breakpoint concentration of the in-vehicle air quality for different types of pollutants has been discussed previously in Table 1. There are various versions of standards and guidelines which depend on the international agencies [22]. Current air quality standards do not provide a breakpoint for CO_2_. So, the CO_2_ breakpoint listed in Table 1 is obtained from different organizations and research groups [16,23,24,25].

Figure 7 shows the flowchart of labeling the in-vehicle air quality index. The parameters of CO_2_, PM_2.5_, and PM_10_ are selected to compute the index for pollutants. The highest index represents the AQI at that time. A large pressure will be created against the vehicle’s body when the vehicle travels at a high speed and leakages will occur between the joints [44]. The higher the vehicle speed, the more outdoor air will be penetrating the vehicle cabin [10]. Thus, other parameters such as temperature, humidity, and vehicle speed may also affect the AQI inside the vehicle cabin.

### 5.3. Normalization

The data normalization method is often implemented in the dataset to ease the data processing time. The function of the normalization is to change the numeric values in the dataset into a 0 to 1 range without changing the original range values of the dataset. In AI prediction, not every dataset requires normalization. However, the sensor dataset contains several features in a different range. For instance, the CO_2_ sensor consisting of three digits might reach to four digits and the PM sensor consisting of one digit might reach three digits. Hence, the min-max scalar is selected due to the sensor dataset in a time-series form with a short interval. If the originality of the trend is not preserved, the learning process in the ML prediction models will be affected.

## 6. Prediction Analysis using Machine Learning Algorithms

This section describes the algorithms that are used to predict the future condition of the air quality.

### 6.1. Linear Regression

A linear regression model is capable of time series prediction [49]. This is because the model makes a prediction by simply computing a weighted sum of the input features, plus a constant called the bias term, as shown in Equation (3).
(3)$$\hat{y}=\theta_{0}+\theta_{1}x_{1}+\theta_{2}x_{2}+\ldots +\theta_{n}x_{n}$$
where, $\hat{y}$—the predicted value*n*—the number of features$x_{n}$—the *n*th feature value$\theta_{j}$—the *j*th model parameter

### 6.2. Support Vector Machine

The support vector machine (SVM) can perform linear or nonlinear classification. Besides that, SVM also supports linear and nonlinear regression applications, known as SVR [50]. In the SVR, there are three parameters that need to be appropriately selected to achieve higher prediction accuracy and better performance. These are the insensitive loss coefficient (ℇ), error penalty factor (C), and kernel function coefficient (γ). The complexity of the model is dependent on these parameters. These three parameters are highly inter-related and affect the SVR model. The grid search method provides the best combination for the three mentioned parameters. By implementing the GridSearchCV function in the sklearn library, the grid search range for both ℇ and C is set (−3, 3, 21) with the logspace function. Thus, the best hyperparameters found for the model of ℇ, C, and γ are at 0.001, 501, and a radial basis function kernel (RBD kernel), respectively.

### 6.3. Multilayer Perceptron

The MLP is the most commonly used model in the feed-forward neural network. The basic MLP has three layers which are the input layer, hidden layer, and output layer as shown in Figure 8. A grid search method is implemented to search the fine-tuned hyperparameters in the MLP. The range of hyperparameters for hidden nodes, learning rate, optimizer, and activation function are 2^3^–2^10^, 0.001–0.05, adam or stochastic gradient descent, and relu or tanh, respectively. The fine-tuned hyperparameters in the MLP structure applied in this research have a single hidden layer of 128 hidden nodes, a 0.001 learning rate, tanh activation function, and stochastic gradient descent optimizer.

### 6.4. Evaluation Methods

To evaluate the performance of the ML applied to the in-vehicle air quality monitoring system, *RMSE* [51], the mean squared error (*MSE*), mean absolute error (*MAE*), and coefficient of determination (*R*^2^) are selected. Equations (4)–(7) present the formula for each of the evaluation metrics, respectively.
(4)$$RMSE=\sqrt{\frac{1}{n}\;\sum_{i=1}^{n}{\left(y_{i}-{\hat{y}}_{i}\right)}^{2}}$$
(5)$$MSE=\frac{1}{n}\sum_{i=1}^{n}{\left(y_{i}-{\hat{y}}_{i}\right)}^{2}$$
(6)$$MAE=\frac{1}{n}\sum_{i=1}^{n}\left|y_{i}-{\hat{y}}_{i}\right|$$
(7)$$R^{2}=1-\frac{\sum_{i=1}^{n}{\left(y_{i}-{\hat{y}}_{i}\right)}^{2}}{\sum_{i=1}^{n}{\left(y_{i}-{\bar{y}}_{i}\right)}^{2}}$$
where, ${\hat{y}}_{i}$—predicted value of *y*${\bar{y}}_{i}$—mean value of *y*

## 7. Results and Discussions

All data were separated into two sets, which are 80% as the training data and 20% as the test data. The model scripts were executed using the Nvidia GeForce RTX 2080 Ti graphics card as the hardware accelerator. 75 data points, which represented data of 5 min, were predicted using the different ML prediction models mentioned in previous sections. There are two datasets to test, train, and evaluate which have different time slots and monthly data as mentioned in Table 2.

The evaluation results of the ML prediction models are shown in Table 3. The evaluation result is to verify the prediction capability. The accuracy of the MLP model had a significant improvement when the historical data was increased from 0.7151 to 0.9107 of *R*^2^. The SVR model with the RBF kernel had the highest *R*^2^ and lowest *MSE*, *RMSE*, and *MAE* compared to other models. The SVR-RBF-based prediction model showed the highest prediction accuracy and had better generalization performance. The *R*^2^ obtained was as high as 0.9890 (section) and 0.9981 (month).

In addition, another important aspect is the computation time of the model for future implementation of the prediction algorithm in edge computing. In the section dataset, the LR had an outstanding computation. It only took 0.2 s for the prediction. However, the *R*^2^ of the LR was lower than the SVR. The prediction model of the SVR had an acceptable computation time (1.6 s) with a high *R*^2^. For the real-time prediction model, high accuracy and low computation time are an important aspect that must be considered.

Figure 9 and Figure 10 show the distributions of prediction results for easy interpretation. From the graph, the prediction model of the SVR-RBF shows similar shapes and tendencies to the actual data. The LR prediction model also shows a good fitting line. However, the MLP prediction model does not fit into the actual data. Hence, the SVR with the RBF prediction model is suitable for system prediction.

## 8. Conclusions

This research focused on the ML prediction model for an in-vehicle air quality application. A hardware testbed was developed to obtain sensor data in the in-vehicle indoor environment. Then, three predictive models of machine learning algorithms such as LR, SVR, and MLP were applied to the in-vehicle air quality prediction system to predict the air quality inside the vehicle cabin. This allowed the monitoring of the real-time air quality inside the car cabin. The system can be used as a potential measure to reduce traffic accidents due to driver drowsiness and fatigue. The results showed that the SVR had the highest performance rates in terms of *R*^2^ and had less error rate. This indicates that the SVR model has an outstanding prediction performance as well as low computation time compared with the LR and MLP models.

## Figures and Tables

**Figure 1 sensors-21-04956-f001:**
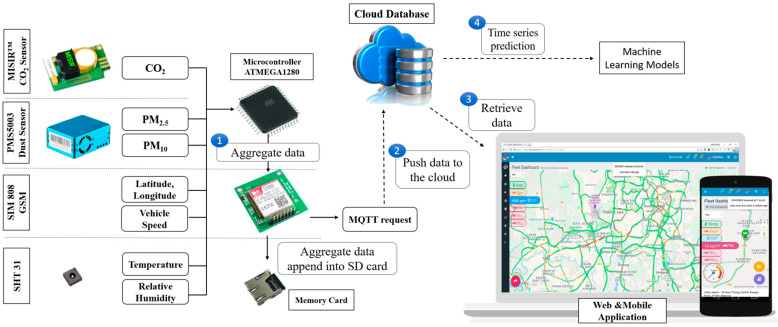
System architecture design for the in-vehicle air quality monitoring system.

**Figure 2 sensors-21-04956-f002:**
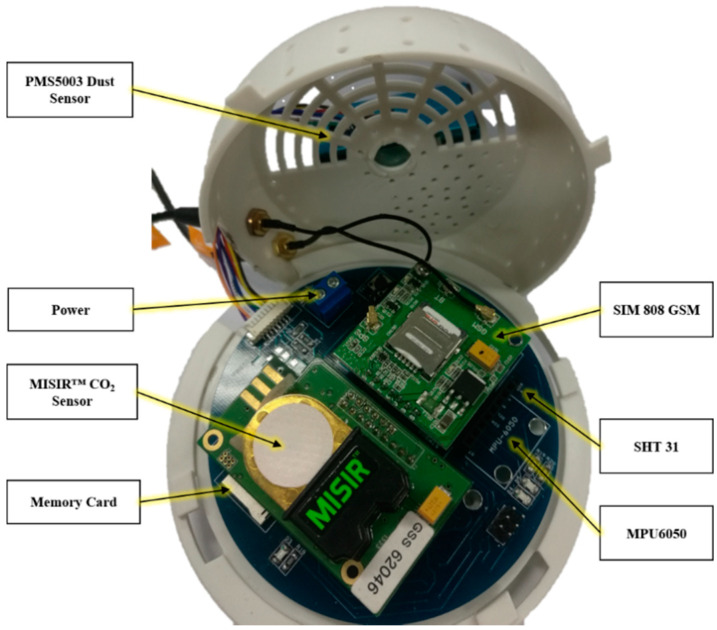
The final design for the device node.

**Figure 3 sensors-21-04956-f003:**
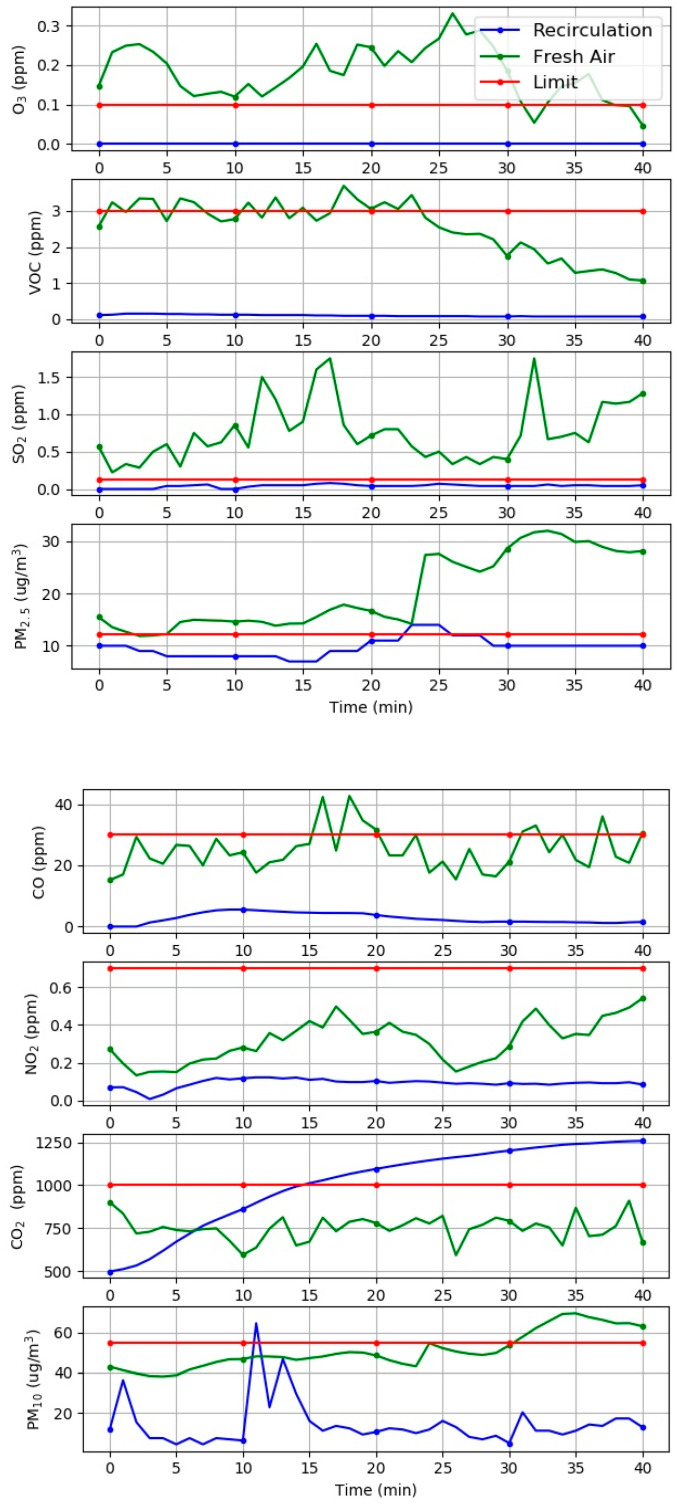
The common hazardous gases inside the vehicle cabin with the FA and RC ventilation modes.

**Figure 4 sensors-21-04956-f004:**
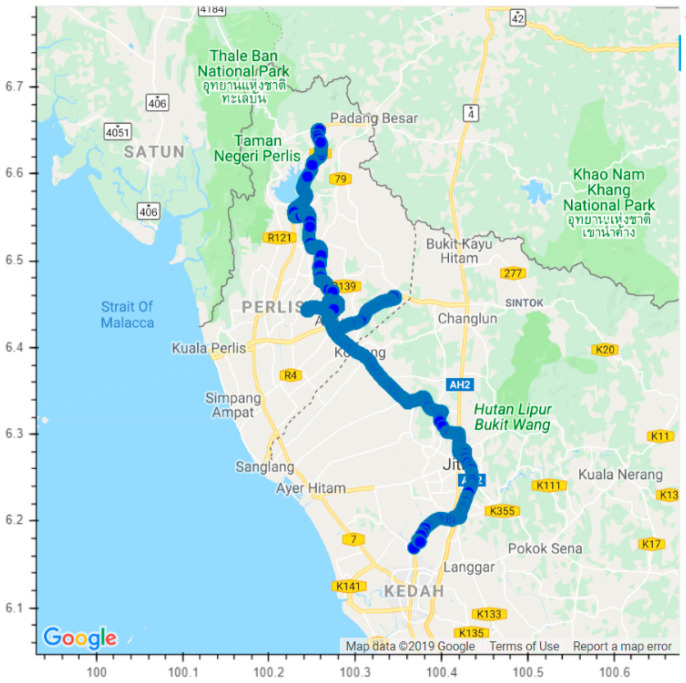
The daily travelled path in the experiment.

**Figure 5 sensors-21-04956-f005:**
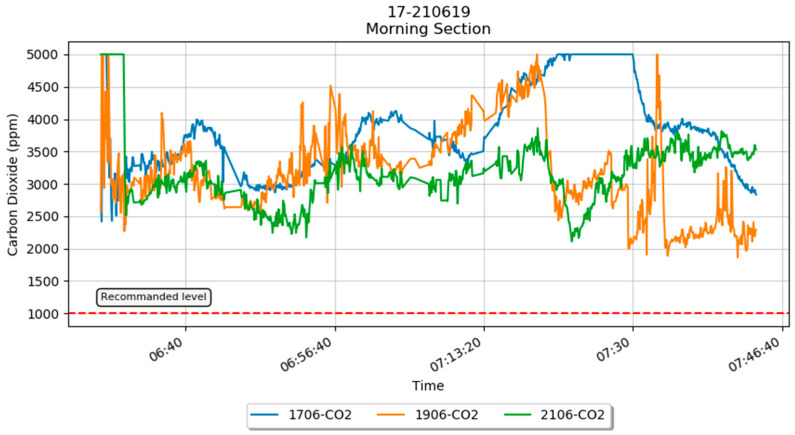
The raw carbon dioxide sensor data plotted for the morning slot.

**Figure 6 sensors-21-04956-f006:**
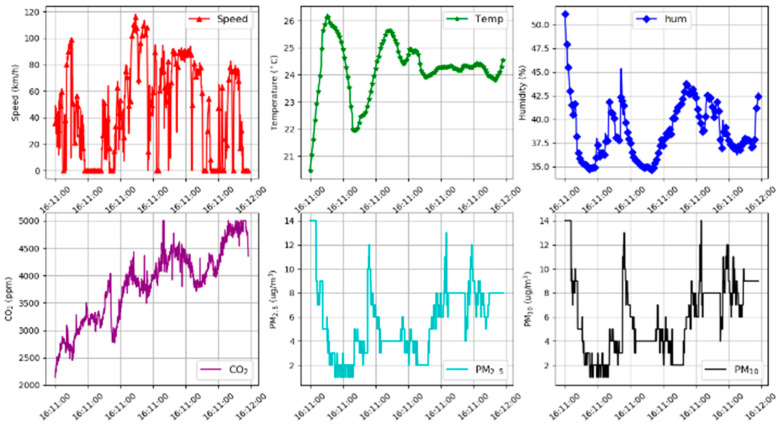
The raw section sensor data plotted for each parameter.

**Figure 7 sensors-21-04956-f007:**
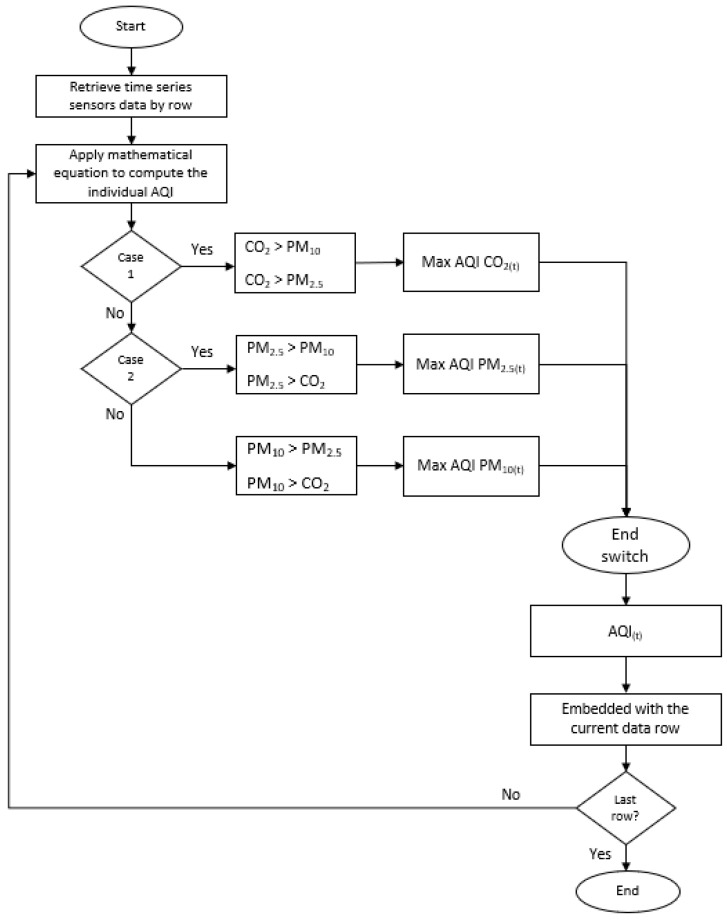
The time series data labelling of the AQI.

**Figure 8 sensors-21-04956-f008:**
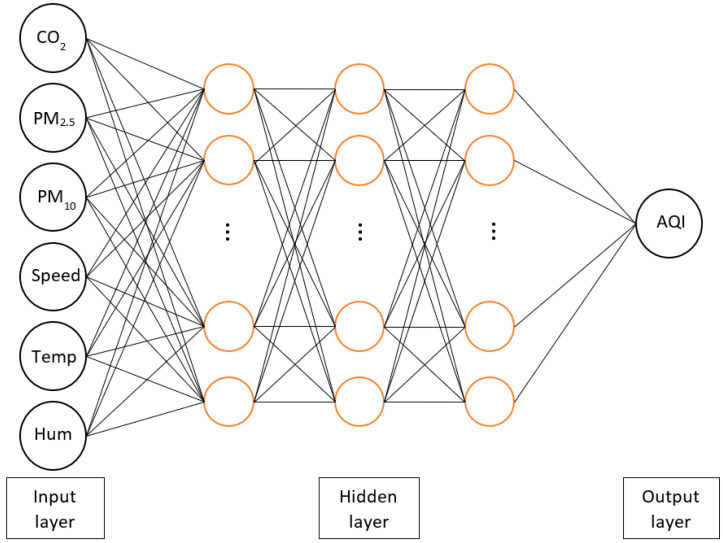
The MLP structure with various hidden node numbers.

**Figure 9 sensors-21-04956-f009:**
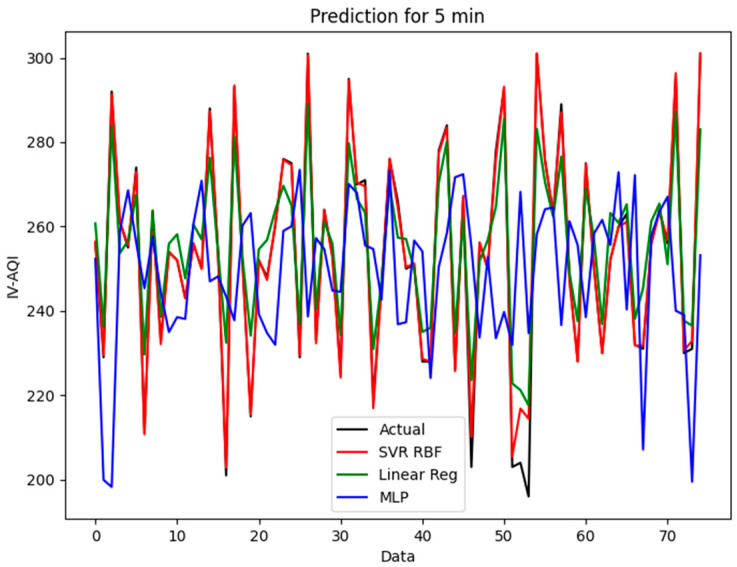
The prediction for the section event in SVR, LR, and MLP.

**Figure 10 sensors-21-04956-f010:**
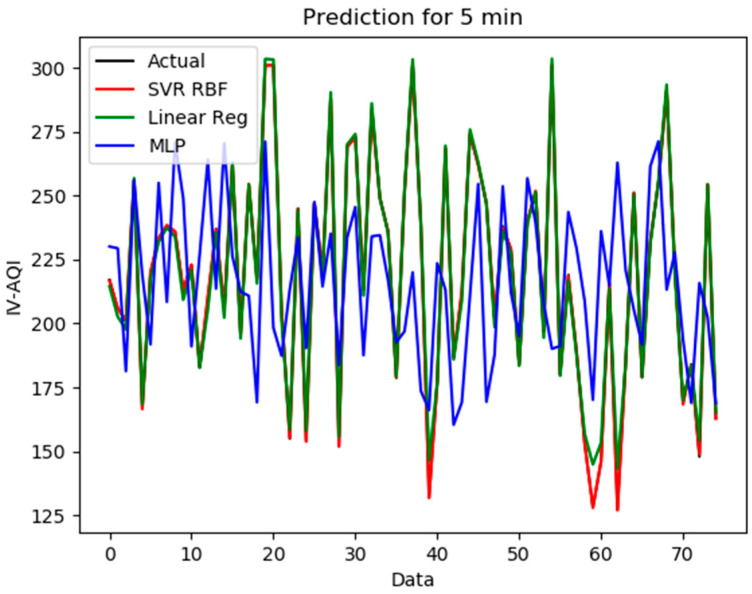
The prediction for the month event in SVR, LR, and MLP.

**Table 1 sensors-21-04956-t001:** Breakpoint concentration of the in-vehicle air quality.

CO_2_ (ppm)	PM_2.5_ (µg/m^3^) ^d^	PM_10_ (µg/m^3^) ^d^	IV-AQI	Five Bands of IV-AQI
C_low_–C_high_	C_low_–C_high_	C_low_–C_high_	I_low_–I_high_	
340–600 ^a^	0.0–12.0	0–54	0–50	Good
601–1000 ^b^	12.1–35.4	55–154	51–100	Moderate
1001–1500	35.5–55.4	155–254	101–150	Unhealthy for sensitive group
1501–2500 ^a^	55.5–150.4	255–354	151–200	Unhealthy
2501–5000 ^c^	150.5–250.4	355–424	201–500	Very unhealthy

a: Associations of the cognitive function scores with carbon dioxide, ventilation, and volatile organic compound exposures in office workers: A controlled exposure study of green and conventional office environments (USA) [16]. b: Industry code of practice on indoor air quality 2010 (Malaysia) [23]. c: Occupational safety and health administration (OSHA): carbon dioxide in workplace atmospheres (US) [24]. d: Environmental protection agency (EPA) [25].

**Table 2 sensors-21-04956-t002:** Data features and samples size for the in-vehicle air quality.

Collection Site		Parameter
Monthly	Number of records	48,816
	Size	3,593,866 bytes (3.59 MB)
One section	Number of records	1184
	Size	92,160 bytes (0.09 MB)
Value types		Twelve air quality variables (Time, latitude, longitude, speed, CO_2_, temperature, humidity, PM_1_, PM_2.5_, PM_10_, count, label)

**Table 3 sensors-21-04956-t003:** Prediction model results.

	Section					Month				
	*R* ^2^	*MSE*	*RMSE*	*MAE*	Computation Time (s)	*R* ^2^	*MSE*	*RMSE*	*MAE*	Computation Time (min)
**SVM**	0.9890	6.4513	2.5410	0.97194	1.6	0.9981	3.6168	1.9018	0.4101	44.5
**LR**	0.8137	109.9008	10.4833	5.1379	0.2	0.9946	10.1875	3.1917	2.1348	37.2
**MLP**	0.7151	212.4807	14.5767	11.5757	26	0.9107	100.0034	9.0589	5.0422	83.3

## Data Availability

All data generated or appeared in this study are available upon requested by contact with the corresponding author. Moreover, the models and code used during the study cannot be shared at this time as the data also forms part of an ongoing study.

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
