# Peer review of "Real-Time In-Vehicle Air Quality Monitoring System Using Machine Learning Prediction Algorithm"

_sensors, 2021, doi:10.3390/s21154956_

Round 1

Reviewer 1 Report

1. There are many writing errors in the article.
2. The author discussed methods of forecasting in "Related Work," but there are too few documents on forecasting methods, and some important references need to be added: "Mu, B., Li, S. and Yuan, S., 2017. Air Quality Forecast Based on Principal Component Analysis-Genetic Algorithm and Back Propagation Model. International Journal of Environmental and Ecological Engineering, 10(8), pp.899-906.", "Distributed Deep Fusion Predictor for a Multi-Sensor System Based on Causality Entropy. Entropy 2021, 23, 219. https://doi.org/10.3390/e23020219", "Mu, B., Li, S. and Yuan, S., 2017. Air Quality Forecast Based on Principal Component Analysis-Genetic Algorithm and Back Propagation Model. International Journal of Environmental and Ecological Engineering, 10(8), pp.899-906", "The New Trend of State Estimation: From Model-Driven to Hybrid-Driven Methods. Sensors 2021, 21, 2085. https://doi.org/10.3390/s21062085"
3. The data size shown in Table 2 includes data such as time, longitude, and latitude, and the meaning of PM1 and other data used for prediction should be explained in detail.
4. How to set the required parameters for SVR application grid search?
5. Why not use the grid search mentioned in SVR to select MLP parameters?
6. The MLP method explains the normalization of the data. Is the normalization also performed when using other methods for prediction? In addition, the normalization process should be explained in the data preprocessing section.
7. This article only uses three simple machine learning methods to make predictions and compare them. Is the learning and prediction of large amounts of data accurate? Have you considered deep learning algorithms?
8. The data in Table 3 should be further analyzed and explained: Why does the calculation time of MLP is the least? Because, in general, MLP will cost more time than SVM.
9. A format error appears on line 386.

Reviewer 2 Report

The paper presents a study of indoor vehicle air quality based on six parameters taken from five sensors for the air and vehicle speed. They compared three ML techniques.

Some minor errors should be corrected, like some references are not correctly cited (for example, lines 35, 38, 51, 215). Also as an example the next lines require syntax verification:

Line 75 should say “which are the best predictive models…”

Line 89 “…an environment with…”

Line 329 “…a weighted sum of the input…”

And RMSE is first used in line 161, and until line 359 the authors indicate what does it stand for.

I recommend to remove Figure 8, it does not provide any information, it is the regular data process in ML.

More important the asseveration in line 38 and 39 “Previous…” requires citation, and not all the accident have that cause. Also, the one in line 90 requires citation.

In the introduction it is not clear, what is the advantage or difference between the authors’ proposal and the one in Hable-Khanderkar & Srinath, 2018. It is very important to make it clear why this study is different from others.

Finally, related to the aforementioned, results compared with other studies using these ML techniques should be analyzed.

The paper presents a study of indoor vehicle air quality based on six parameters taken from five sensors for the air and vehicle speed. They compared three ML techniques.

Some minor errors should be corrected, like some references are not correctly cited (for example, lines 35, 38, 51, 215). Also as an example the next lines require syntax verification:

Line 75 should say “which are the best predictive models…”

Line 89 “…an environment with…”

Line 329 “…a weighted sum of the input…”

And RMSE is first used in line 161, and until line 359 the authors indicate what does it stand for.

I recommend to remove Figure 8, it does not provide any information, it is the regular data process in ML.

More important the asseveration in line 38 and 39 “Previous…” requires citation, and not all the accident have that cause. Also, the one in line 90 requires citation.

In the introduction it is not clear, what is the advantage or difference between the authors’ proposal and the one in Hable-Khanderkar & Srinath, 2018. It is very important to make it clear why this study is different from others.

Finally, related to the aforementioned, results compared with other studies using these ML techniques should be analyzed.

Reviewer 3 Report

The main concern is about novelty. Systems like the proposed already exist and it is not clear major contribution

State of art should identify related works and identify the research gap.

Figure 1 does not belong to authors and I do not know if they have the authorization to use

Comparison with data collected from vehicles (moving) and static sensors should be discuss

Algorithms should have more details about the implementation

The conclusion again is not clear major contribution. 

Round 2

Reviewer 1 Report

The reviewer carefully read the revised manuscript and believes that the author has adequately explained the reviewer's comments and made detailed revisions to the paper. The paper has been greatly improved. Finally, there is a small comment: Fig.9 is not as clear as Fig.10.

Author Response

Point 1: Fig.9 is not as clear as Fig.10.

Response 1: Thank you for the comment. Resolution of Figure 9 has been improved.

Reviewer 3 Report

Authors should explain better the difference between these measurements and city air quality measurements already performed and analysed. Air quality major problems are in the cities...

Parts need clarification

What is predictive analytics that enable visualization? I do not understand if I think this does not make any sense

This can be used as a potential measure to reduce traffic accident due to the driver’s drowsiness and fatigue (this at a conclusion level maybe used at abstract does not make sense - and all contribution should be validaded in authors work)

line 260 ML will be generated by implementing the data preprocessing algorithm that will further discuss in section 5.1  (this part is about data prepararion - does not make sense and all section 5 does not provide details about the implementation)
